# AgNP Composite Silicone-Based Polymer Self-Healing Antifouling Coatings

**DOI:** 10.3390/ma17174289

**Published:** 2024-08-30

**Authors:** Xingda Liu, Jiawen Sun, Jizhou Duan, Kunyan Sui, Xiaofan Zhai, Xia Zhao

**Affiliations:** 1State Key Laboratory of Bio-Fibers and Eco-Textiles, College of Materials Science and Engineering, Shandong Collaborative Innovation Center of Marine Biobased Fibers and Ecological Textiles, Qingdao University, Qingdao 266071, China; 2021020485@qdu.edu.cn (X.L.); sky@qdu.edu.cn (K.S.); 2Key Laboratory of Advanced Marine Materials, Key Laboratory of Marine Environmental Corrosion and Bio-Fouling, Institute of Oceanology, Chinese Academy of Sciences, Qingdao 266071, China; duanjz@qdio.ac.cn (J.D.); zhaixf@qdio.ac.cn (X.Z.); zx@qdio.ac.cn (X.Z.)

**Keywords:** antifouling coating, self-healing, hydrogen bonding, imine linkage, AgNPs

## Abstract

Biofouling poses a significant challenge to the marine industry, and silicone anti-biofouling coatings have garnered extensive attention owing to their environmental friendliness and low surface energy. However, their widespread application is hindered by their low substrate adhesion and weak static antifouling capabilities. In this study, a novel silicone polymer polydimethylsiloxane (PDMS)-based poly(urea-thiourea-imine) (PDMS-PUTI) was synthesized via stepwise reactions of aminopropyl-terminated polydimethylsiloxane (APT-PDMS) with isophorone diisocyanate (IPDI), isophthalaldehyde (IPAL), and carbon disulfide (CS_2_). Subsequently, a nanocomposite coating (AgNPs-x/PDMS-PUTI) was prepared by adding silver nanoparticles (AgNPs) to the polymer PDMS-PUTI. The dynamic multiple hydrogen bonds formed between urea and thiourea linkages, along with dynamic imine bonds in the polymer network, endowed the coating with outstanding self-healing properties, enabling complete scratch healing within 10 min at room temperature. Moreover, uniformly dispersed AgNPs not only reduced the surface energy of the coating but also significantly enhanced its antifouling performance. The antibacterial efficiency against common marine bacteria *Pseudomonas aeruginosa* (*P*.sp) and *Staphylococcus aureus* (*S*.sp) was reduced by 97.08% and 96.71%, respectively, whilst the diatom settlement density on the coating surface was as low as approximately 59 ± 3 diatom cells/mm^2^. This study presents a novel approach to developing high-performance silicone antifouling coatings.

## 1. Introduction

Marine biofouling refers to the adherence and accumulation of microorganisms such as bacteria, diatoms, and larger organisms like barnacles, mussels, oysters, and tubeworms on the surfaces of submerged artificial structures. It is associated with detrimental effects, including microbial corrosion, ship drag, and biological invasions, thus posing substantial safety hazards to various underwater facilities and equipment.

For over a century, antifouling coatings have been an effective solution to fouling. While tributyltin (TBT) antifouling coatings exhibit excellent antifouling performance, they release organotin biocides into seawater, severely affecting the survival of other marine organisms, which led to a complete ban on their use in 2008 [1,2]. Self-polishing coatings (SPC) have emerged as an effective type of antifouling coating in recent years [3]. Their antifouling mechanism involves the hydrolysis of resin in water, which then slowly releases low-toxicity antifouling agents such as cuprous oxide. However, microplastics generated during resin hydrolysis may pose potential adverse effects on the marine environment [4]. In contrast, PDMS-based fouling-release coatings have recently developed rapidly owing to their environmental friendliness. The low surface free energy and elastic modulus make it difficult for fouling organisms to adhere to the surface or allow them to be easily removed by the sheer force of water flow. However, PDMS surfaces cannot remove the mucus secreted by bacteria and diatoms at low water flow rates, leading to poor static antifouling capability. Moreover, PDMS is an elastomer that generally suffers from low adhesion and susceptibility to damage, limiting its widespread application [5].

To address the aforementioned shortcomings, studies have focused on the addition of antifouling agents, the incorporation of nanofillers, and the endowment of self-healing properties to coatings. Xiong et al. [6] fabricated various LAP/PDMS composite antifouling coatings by introducing long afterglow phosphors (LAP) with different emission intensities of blue-green (BG), yellow-green (YG), and sky blue (SB) into PDMS. These coatings emit light in dark conditions, disrupting the physiological activities of marine fouling organisms surrounding the coatings, thereby improving their antifouling performance. Selim et al. [7] manufactured polydimethylsiloxane (PDMS)/ZnO nanorod (NR) composites using an in-situ method to explore the effect of ZnO NR on the superhydrophobicity and antifouling properties of silicone coatings. At a ZnO NR content of 0.5 wt%, it exhibited favorable dispersion abilities, thus enhancing the hydrophobicity, self-cleaning, and antifouling properties of the composite. To enhance coating durability, researchers have also developed self-healing antifouling coatings by constructing dynamic multiple hydrogen bond interactions between hydrogen donors and highly electronegative atoms (hydrogen acceptors) on various polymer chains, forming reversible cross-links. In our previous works [8,9], long-lasting antifouling coatings with efficient self-healing capabilities were prepared through multiple dynamic hydrogen bond interactions between urea and thiourea linkages, as well as disulfide bonds. In addition, the introduction of lipoic acid-benzothiazole antibacterial groups optimized the antifouling performance. Marine field tests demonstrated that this coating possesses long-term static antifouling capabilities. Additionally, imine bonds, also termed Schiff base bonds, are another type of dynamic covalent bond formed by the reversible condensation of primary amines and aldehydes. Once disrupted, dynamic imine bonds can spontaneously re-form, endowing the polymer with self-healing properties [10].

Silver nanoparticles (AgNPs) have been universally recognized for their efficient broad-spectrum antibacterial properties [11,12,13]. The positively charged silver ions are attracted to the negatively charged microbial cell membranes, enabling AgNPs to penetrate cells, disrupt cellular molecules, and cause intracellular damage. Subsequently, AgNPs induce the formation of reactive oxygen species (ROS) in bacterial cells, leading to the dephosphorylation of tyrosine, which hinders cell signal transduction pathways [14,15,16,17]. Liu et al. [18] constructed self-healing and long-lasting antifouling AgNP hybrid silicone coatings by synthesizing coordination polymers with Cu^2^^+^ as the coordinating metal and polydimethylsiloxane (PDMS) and polytetramethylene glycol (PTMG) as the main chains and introducing AgNPs. The excellent antibacterial activity of AgNPs imparted the coating with superior antibacterial properties and the ability to prevent bacterial biofilm formation. Cui et al. [19] chemically bonded silver nanoparticles with silver, copper, and zinc ternary ion-exchanged zeolites through α-lipoic acid and encapsulated them in a hydrophilic polymer to generate a highly antibacterial, highly active, and durable silver ion nanocomposite antibacterial powder coating additive. The formed silver nanoparticle thin layer and hydrophilic film extended the release of active Ag^+^ from the zeolite, whilst Ag^+^ promoted the activation of AgNPs, thereby exerting satisfactory antibacterial effects.

In this study, aminopropyl-terminated polydimethylsiloxane (APT-PDMS) was utilized as the main chain and reacted with isophorone diisocyanate (IPDI), isophthalaldehyde (IPAL), and carbon disulfide (CS_2_) to synthesize a PDMS-based polyurea-thiourea-imine polymer (PDMS-PUTI). The multiple dynamic hydrogen bonds and imine bonds in PDMS-PUTI synergistically conferred excellent self-healing ability on the coating, allowing self-healing at room temperature. Subsequently, PDMS-PUTI was combined with AgNPs to yield the nanocomposite polymer coating (AgNPs-x/PDMS-PUTI). The broad-spectrum and highly efficient antibacterial and anti-algal capabilities of AgNPs endowed the coating with superior antifouling properties. Overall, this study aimed to develop an environmentally friendly composite coating for marine anti-biofouling.

## 2. Materials and Methods

### 2.1. Materials

APT-PDMS (*M*_w_ = 2500 g·mol^−1^) was purchased from Silong Material Technology Co., Ltd. (Hangzhou, China). Sylgard 184 PDMS was procured from the Dow Corning Corporation (Midland, MI, USA). Tetrahydrofuran (THF, 99.5%), IPDI (99%), IPAL (98%), CS_2_ (99.9%), and dichloromethane (DIM, 99.9%) were sourced from Macklin. Silver nanoparticles (AgNPs, <100 nm particle size) and anhydrous ethanol were acquired from Aladdin and used without further treatment. Unless otherwise specified, all additional reagents were utilized in the form supplied by the manufacturer.

### 2.2. Preparation of AgNPs-x/PDMS-PUTI

As illustrated in Figure 1, 0.41 g of IPDI was dissolved in 10 mL of THF and transferred to a nitrogen-filled three-neck flask for protection. Then, 7.23 g of APT-PDMS was dissolved in 20 mL of THF and placed in a constant pressure dropping funnel, which was added dropwise into the three-neck flask. After stirring for 30 min, 0.14 g of IPAL was added. The mixture was stirred for 3 h, following which the temperature was raised to 60 °C. After 1 h of reaction at this temperature, the solution was cooled to 45 °C, and 1.53 g of CS_2_ was added, and the mixture was allowed to react for 3 h. The final product was then subjected to rotary evaporation and vacuum drying to obtain PDMS-PUTI. An amount of 10 g of the PDMS-PUTI was subsequently dissolved in THF, and AgNPs (0.3 g, 3 wt%) were added. Finally, the mixture was poured into a polytetrafluoroethylene (PTFE) mold and vacuum dried. All prepared coatings were named AgNPs-X%/PDMS-PUTI, and X% represents the mass ratio of AgNPs.

### 2.3. Characterization

^1^H NMR was recorded using a Bruker Avance 500 MHz NMR spectrometer (Billerica, MA, USA) at 25 °C using deuterated chloroform (CDCl_3_) as the solvent and tetramethylsilane (TMS, δ = 0) as the internal standard. GPC was carried out on Agilent 1260 Infinity (Santa Clara, CA, USA) equipped with refractive index (RI) and ultraviolet (UV) detectors. The column used was Agilent PLgel 5 μm MIXED-C, with THF serving as the eluent at a flow rate of 1 mL min^−1^. Calibration was performed via pretests on polystyrene standards (1.30 × 10^3^ to 2.21 × 10^6^ g mol^−1^) with narrow polydispersity, and the number- and weight-average molecular weights (*M*_n_ and *M*_w_, respectively) and the ratio (the polydispersity index, PDI = Mn/Mw) of polymers were estimated. Fourier-transform infrared spectroscopy (FTIR) was conducted on a Bruker TENSOR-27 infrared spectrophotometer using the KBr disc method. Each spectrum comprised 32 scans between 4000 and 500 cm^−1^. Using a universal testing machine (Model Z020, ZwickRoell, Ulm, Germany) at a rate of 150 mm/min, tensile tests were conducted on dumbbell-shaped specimens measuring 75 mm by 12.5 mm by 4 mm. Fracture surfaces and energy-dispersive spectroscopy (EDS) elemental maps of AgNPs-x/PDMS-PUTI were obtained using a scanning electron microscope (SEM, S-4800, Hitachi, Tokyo, Japan).

### 2.4. Self-Healing Ability

Self-healing abilities were evaluated at room temperature by completely scratching the coating using a scalpel under ambient air conditions. Afterward, the coated samples were inverted on the sample stage, and the healing process was monitored using a microscope. Snapshots were captured every minute to visualize the healing process.

### 2.5. Adhesion Test

The degree of adhesion was measured according to ASTM-D4541 standards [20]. Briefly, five cylindrical aluminum dollies (diameter: 20 mm) were bonded onto glassfiber-reinforced epoxy (GFE) and Q235 steel using epoxy adhesive (Araldite). A pull-off adhesion tester (PosiTest AT-A Automatic, Defelsko, New York, NY, USA) was used to automatically measure the adhesive strength of each sample.

### 2.6. Surface Performance

A laser confocal microscope was used to characterize surface roughness over a 367 μm × 367 μm area. The static water contact angle (WCA) and diiodomethane (DIM) contact angle of all samples were determined at 25 °C using a contact angle goniometer (OCA 25, Dataphysics, Filderstadt, Germany). Each sample underwent five measurements, and the average value was recorded. The surface free energy of samples was calculated using the Owens–Wendt–Rabel–Kaelble equation based on the measured contact angles [21]. The pseudobarnacle removal strength test was conducted according to standard procedures. Under ambient conditions, five cylindrical aluminum pins (10 mm in diameter and height) were bonded to different areas of the coating surface using a two-component epoxy adhesive (Ergo 5800, Kisling, Zurich, Switzerland) and cured for 3 days. Subsequently, a thrust gauge (SHSIWI, SH-500, Shanghai, China) was employed to apply a shear force parallel to the surface of the aluminum pillars to evaluate the fouling release performance by determining pseudobarnacle adhesion strength [22]. 

### 2.7. Antibacterial and Antidiatom Settlement Tests

*Pseudomonas* sp. (*P*.sp.) and *Staphylococcus aureus* (*S*.sp.) were used to assess the antibacterial performance of the samples. *P*.sp. and *S. aureus* were initially cultured in liquid LB medium at 37 °C for 10 h. Thereafter, the bacteria were diluted and suspended in 0.1 M PBS buffer. Antibacterial experiments were conducted using PBS medium containing 10^7^ cfu/mL of *P*.sp. and *S. aureus*, respectively. Bare glass slides (control), PDMS elastomer (Sylgard 184) (reference), and AgNPs-x/PDMS-PUTI were sliced into 1 cm × 1 cm pieces and sterilized under UV light for 30 min. Each sample was then immersed in the prepared bacterial suspension and incubated at 25 °C for 6 h. Next, the samples were stained with the LIVE/DEAD BacLight Bacterial Viability Kit for 15 min and rinsed twice with PBS buffer to discard weakly adhered bacteria and excess dye. The attached bacteria were observed under a fluorescence microscope (Olympus BX-51, Tokyo, Japan), and five fields of view were counted for each of the three parallel samples. The antibacterial adhesion rate (R) was calculated as R = (Nc − Ns)/Nc, where Nc and Ns represent the bacterial count on the control and coated glass slides, respectively. All media were sterilized at 121 °C for 30 min prior to use. Inoculation and immersion experiments were carried out in an AIRTECH clean bench after 30 min of UV irradiation [23]. The number of bacteria in each field of view was counted using Image J software (https://imagej.net/ij/download.html, accessed on 22 August 2024) and compared to the bacteria count on a blank glass plate to determine the antibacterial rate for different images. The average value was used as the antibacterial rate for the sample, and the error bars represent the range of antibacterial rates across different fields of view.

Navicula incerta was used as a model diatom, sourced from the Qingdao National Laboratory for Marine Science and Technology. Diatom settlement tests were conducted on uncoated glass slides (control), PDMS elastomer (Sylgard 184) (reference), and AgNPs-x/PDMS-PUTI. Each sample was placed in a six-well plate with 10 mL of diatom cell suspension (1.2 × 10^5^ cells/mL) in each well. The samples were maintained at 23 °C for 24 h (12 h light, 12 h dark), then rinsed with sterile seawater to discard unattached diatoms. The attached diatoms were observed under a fluorescence microscope (Olympus BX-51, Japan), and five fields of view were counted for each of the three parallel samples using Image J software.

## 3. Results

### 3.1. Structural Characterization

Figure 2 displays the ^1^H NMR spectrum of AgNPs-x/PDMS-PUTI. The polymer is a linear polymer with Si-O bonds as the main chain, containing urea group, imine bonds, and thiourea group, where multiple strong cross-linking hydrogen bonds are formed between the polyurea and thiourea bonds, cross-linking into an elastomer at room temperature. The proton signals of the urea linkages and adjacent methylene groups were located at 3.60 (10), 3.03 (11), and 3.14 (8) ppm, respectively. At the same time, the peak at 3.83 (16) corresponded to the tertiary carbon adjacent to the urea group (3.80 ppm). Imine-related proton signals were noted at 8.3 (1) ppm, whereas signals assigned to the phenyl groups (three peaks) were identified at 7.60–8.05 (2) ppm [24]. Additionally, a signal related to methylene adjacent to the imine group was detected at 3.65 (4) ppm, and a broad signal for the secondary amine was observed at 4–6 ppm [25]. The characteristic peaks of urea and imine groups in PDMS-PUTI validated the successful synthesis of the polymer with the designed structure.

The polymer structure was further confirmed using FTIR spectra. As depicted in Figure 3, the C=O stretching vibration occurred at 1630 cm^−1^, whilst the secondary amine N-H stretching and bending vibrations were detected at 3338 and 1570 cm^−1^, respectively, reflecting the formation of urea groups [9]. The peak at 1445 cm^−1^ corresponds to the stretching of N-C=S. Furthermore, due to the significant molar ratio of IPDI to IPAL, the characteristic absorption peak of the C=N bond at 1650 cm^−1^ is masked by the absorption peak of the C=O bond at 1630 cm^−1^. This overlap results in a less distinct C=N absorption peak at 1650 cm^−1^. Both FTIR and ^1^H NMR collectively confirmed the chemical structure of PDMS-PUTI. Additionally, the GPC analysis revealed that the molecular weight of the fabricated PDMS-PUTI was 21.7 kDa, with a polydispersity index (PDI) of 3.77 (Figure 4). PDMS-PUTI is a pale yellow transparent elastomer, which turns black-gray after the addition of AgNPs. As shown in Figure 5b, the tensile strength of PDMS-PUTI is 0.72 MPa, while the tensile strengths of AgNPs-3/PDMS-PUTI, AgNPs-6/PDMS-PUTI, and AgNPs-9/PDMS-PUTI are 0.77 MPa, 0.700 MPa, and 0.65 MPa, respectively. This could be attributed to the fact that AgNPs act as a mechanical reinforcement material when present in smaller amounts. However, as the AgNP content increases, agglomeration and sedimentation may occur, leading to a decrease in the mechanical properties of the coating.

### 3.2. Self-Healing Performance

The results of the self-healing test are delineated in Figure 6. The complete scratch healing times for the PDMS-PUTI and AgNPs-3/PDMS-PUTI, AgNPs-6/PDMS-PUTI, and AgNPs-9/PDMS-PUTI coatings were 9 min, 12 min, 15 min, and 20 min, respectively, demonstrating their robust self-healing capabilities. This could be ascribed to the freely moving polymer chains in PDMS-PUTI facilitating the self-healing process. The multiple hydrogen bonds formed by the urea and thiourea groups and the inherent dynamic imine bonds between polymer chain segments synergistically promoted the healing of the coatings [26,27,28]. However, the self-healing capability of the coatings gradually decreased with an increase in AgNP content. This could be attributed to AgNPs affecting the mobility of polymer chain segments, hindering their rearrangement and consequently delaying the self-healing process [29,30].

### 3.3. Adhesion Properties

Coatings with higher adhesive strengths are less likely to detach or peel off from the substrate surface under external forces, thereby preserving the integrity of the coating and enhancing its durability and damage resistance. Adhesion tests were conducted for all samples, and the results are presented in Figure 7. All coatings exhibited higher adhesion strength on Q235 steel compared to GFE, primarily due to mechanical interlocking and hydrogen bonds between the rough steel surface and hydroxyl groups on the surface of the coating. PDMS, possessing a non-polar structure, exhibited weak adhesion strength (GFE: 0.41 MPa, steel: 0.68 MPa) and was prone to detachment from the substrate. In contrast, the adhesion strength of all sample coatings was significantly higher than that of PDMS (GFE: >1.35 MPa, steel: >2.1 MPa). This enhancement in adhesion can be attributed to the formation of covalent bonds between the urea and thiourea groups within the polymer network and the surface hydroxyl groups of the substrate [31]. At the same time, the adhesion strength of PDMS-PUTI ranged from approximately 1.52 MPa on GFE to 2.32 MPa on steel. Interestingly, the adhesion strength of composite coatings progressively decreased to approximately 1.35 MPa on GFE and 2.12 MPa on steel with an increase in AgNP content. Despite a marginal decrease in adhesion strength with increased silver nanoparticle content, the minimum adhesion strength remained at 2.15 MPa, which is sufficient for use in marine environments.

### 3.4. Surface Properties

Furthermore, the surface morphologies of the coatings were examined. As shown in Figure 8, the surface morphology of AgNPs-x/PDMS-PUTI at 5000× magnification was tested via SEM. The SEM images indicate that the surface of PDMS-PUTI is the smoothest, while EDS analysis reveals the absence of Cu and Ag atoms on the coating surface. The surface morphology (Figure 8b–d) indicated uniform dispersion of AgNPs-3/PDMS-PUTI and AgNPs-6/PDMS-PUTI, while AgNPs-9/PDMS-PUTI exhibited agglomeration. As anticipated, the antifouling effect of the coating relied on the uniform dispersion of AgNPs on the polymer surface. In addition, the distribution of surface elements was analyzed using EDS, exposing clear distributions of C, S, Si, Cu, and Ag and corroborating the uniform dispersion of AgNPs. The EDS point scan visually displayed the presence and relative content of C, Si, S, Cu, and Ag on the surface of the AgNPs-x/PDMS-PUTI coating. As illustrated in the EDS images for Cu and Ag in Figure 8, the coating contains almost no copper element, and the content of Ag element increases with the addition of AgNPs.

Surface roughness is also a crucial factor affecting the antifouling performance of coatings. Herein, CLSM was employed to further the surface roughness of the coatings (Figure 9). Compared to PDMS-PUTI, the surface roughness of AgNPs-x/PDMS-PUTI increased with an increase in AgNP content. Specifically, the Sa values for PDMS-PUTI, AgNPs-3/PDMS-PUTI, and AgNPs-6/PDMS-PUTI were 0.105 μm, 0.187 μm, and 0.266 μm, respectively. Indeed, the AgNPs-9/PDMS-PUTI coating exhibited the highest roughness, with a surface arithmetic mean height (Sa) of 0.289 μm. Although AgNPs increased the roughness of the coatings, the composite coatings maintained relatively low surface roughness. Overall, all samples remained relatively smooth, which is beneficial for preventing the attachment of fouling microorganisms to the coating surface.

As shown in Figure 10, the water contact angle (WCA) and surface energy (SE) of the coatings were measured. PDMS-PUTI exhibited good hydrophobic properties and low surface energy. Importantly, the incorporation of AgNPs, a hydrophobic nanomaterial, enhanced the hydrophobic properties of the coatings. Increasing the AgNP content resulted in an increase in the WCA of the coating samples, from 123.46° to 125.67°. Notably, the surface energy of all samples ranged from 21.92 mJ/m^2^ to 23.67 mJ/m^2^. Lastly, the low SE contributed to the effective removal of simulated barnacles from the samples [32,33], with removal strengths ranging between 0.241 MPa and 0.278 MPa for all samples (Figure 11).

### 3.5. Antibacterial and Anti-Diatom Properties

As depicted in Figure 12, PDMS exhibited moderate inhibition of bacterial settlement for *P*.sp. and *S*.sp., with inhibitory rates of approximately 61.67% and 64.43%, respectively. In contrast, PDMS-PUTI, AgNPs-3/PDMS-PUTI, AgNPs-6/PDMS-PUTI, and AgNPs-9/PDMS-PUTI had higher antimicrobial rates against *P*.sp. (roughly 84.27%, 89.17%, 94.07%, and 97.01%, respectively) and *S*.sp. (approximately 90.46%, 92.96%, 96.21%, and 96.71%, respectively). Compared to blank samples and PDMS surfaces, all coating surfaces exhibit reduced bacterial adhesion owing to the effective fouling resistance of AgNPs, with antimicrobial efficacy increasing with increasing AgNP content [34]. Among them, AgNPs-9/PDMS-PUTI had the least degree of bacterial adhesion. The surface energy of PDMS-PUTI is lower, making it easier for bacteria to detach. Additionally, the thiourea groups themselves have certain antimicrobial properties, giving PDMS-PUTI better antibacterial capabilities. Furthermore, AgNPs also slightly reduced the surface energy of the coating. The synergistic effect of AgNPs’ excellent antimicrobial properties and the low surface energy performance of the polymer resulted in the coating having better antifouling properties than PDMS-PUTI. Figure 12c displays fluorescent images of samples immersed in a diatom culture medium for one day. The diatom density on PDMS, PDMS-PUTI, AgNPs-3/PDMS-PUTI, AgNPs-6/PDMS-PUTI, and AgNPs-9/PDMS-PUTI coating surfaces was 483 cell/mm^2^, 221 cell/mm^2^, 135 cell/mm^2^, 64 cell/mm^2^, and 59 cell/mm^2^, respectively. This gradual reduction in diatom density could be attributed to increasing AgNP content, which induces chloroplast degradation and reduces cellular photosynthetic activity. The abundant presence of AgNPs on the coating surfaces inhibited diatom growth, thereby inhibiting diatoms from adhering to the coating surface. 

## 4. Conclusions

In this study, a nanocomposite polymer coating with exceptional self-healing and antifouling properties was prepared by incorporating AgNPs into a composite PDMS-PUTI matrix. Among the formulations, AgNPs-9/PDMS-PUTI exhibited the best antibacterial and anti-algal performance. Moreover, the synergistic effect of multiple hydrogen bonds formed by the urea and thiourea groups, as well as the dynamic covalent imine bonds, conferred the composite coating with rapid self-healing capabilities. Furthermore, the incorporation of AgNPs as nanofillers into the polymer, combined with the low surface energy characteristics of silicone and the antibacterial properties of AgNPs, further optimized the fouling release and antifouling performance of AgNPs-x/PDMS-PUTI coatings. However, with the increase of AgNP content, the adhesion of the coating decreased, the roughness increased, and the self-healing ability weakened. In conclusion, AgNPS-9/PDMS-PUTI had better comprehensive properties. The cost aspect of AgNPs is significant, but by combining them with high-performance polymer matrices to create composite coatings, the amount of AgNPs can be minimized while maximizing their excellent antibacterial properties. This approach holds promise for achieving high antibacterial rates with a lower amount of AgNPs, leading to cost-effective nanocomposite polymer coatings in the future. As a low surface energy antifouling coating, this coating can be applied to various ships, effectively reducing the adhesion of marine organisms, thereby decreasing energy consumption and minimizing the risk of biological invasion. This work provides a promising pathway towards the development of high-performance silicone-based coatings for marine anti-biofouling.

## Figures and Tables

**Figure 1 materials-17-04289-f001:**
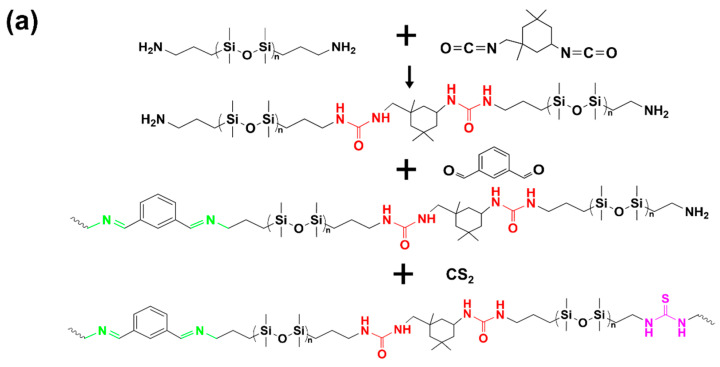
(**a**) Synthesis of AgNPs-X/PDMS-PUTI. (**b**) AgNPs-X/PDMS-PUTI design diagram.

**Figure 2 materials-17-04289-f002:**
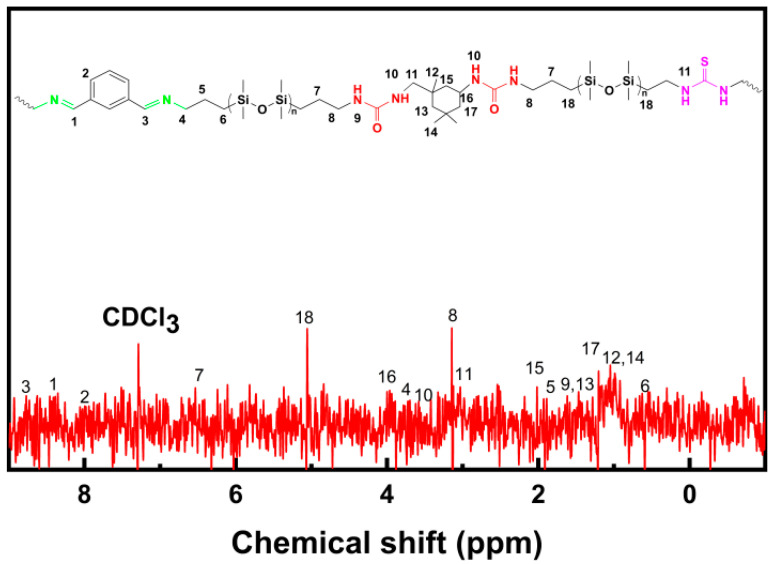
^1^H NMR spectra of PDMS-PUTI.

**Figure 3 materials-17-04289-f003:**
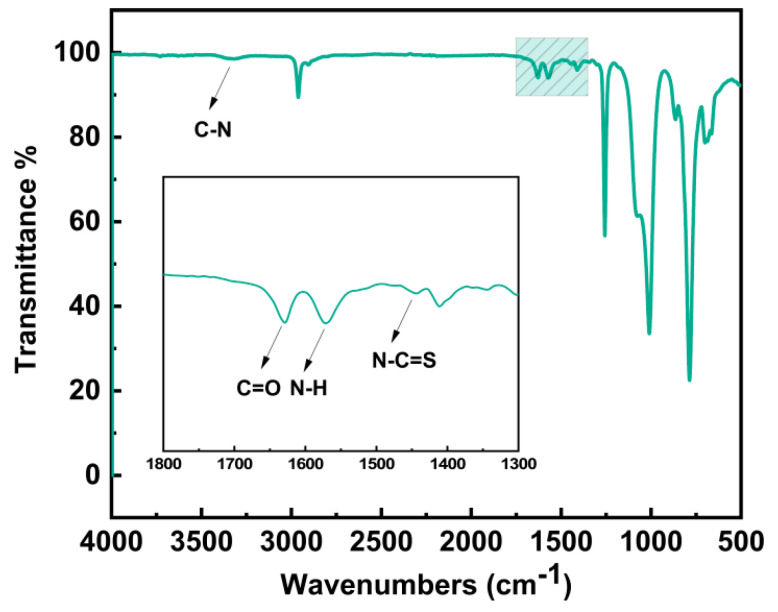
FTIR spectra of PDMS-PUTI.

**Figure 4 materials-17-04289-f004:**
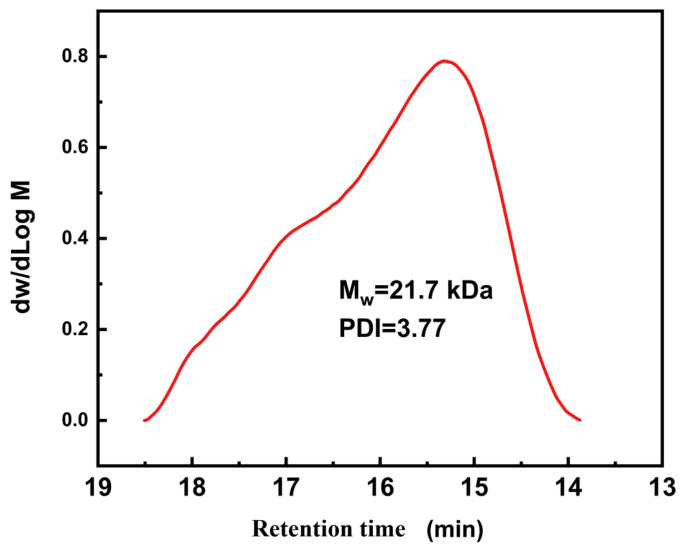
GPC curve of PDMS-PUTI.

**Figure 5 materials-17-04289-f005:**
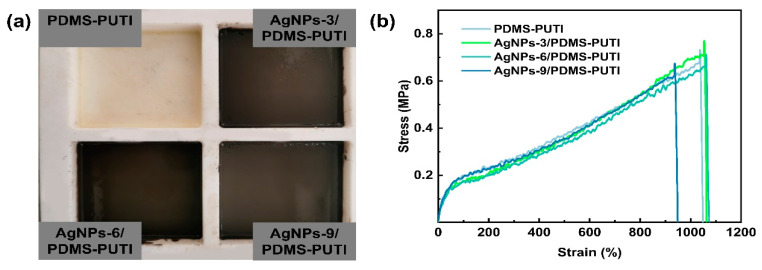
(**a**) Appearance of PDMS-PUTI and AgNPs-x/PDMS-PUTI. (**b**) The stress–strain curves at 25 °C of PDMS-PUTI and AgNPs-x/PDMS-PUTI.

**Figure 6 materials-17-04289-f006:**
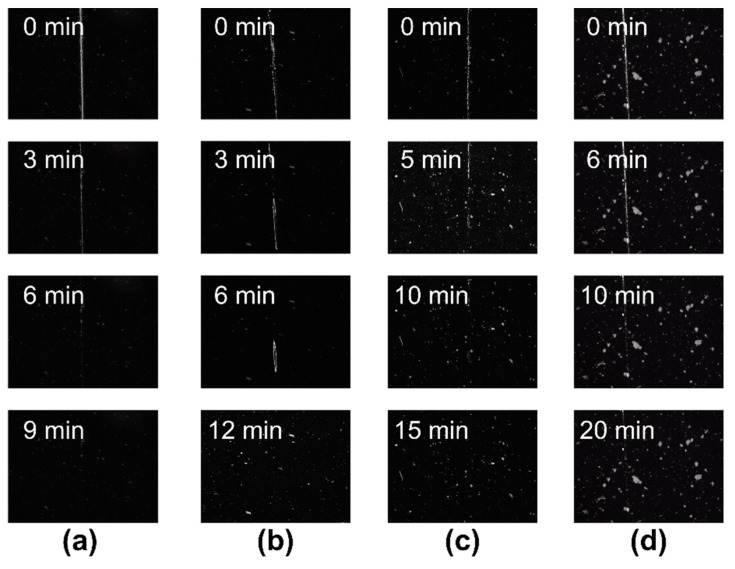
Self-healing properties of the AgNPs-x/PDMS-PUTI coatings. Micrographs images of the self-healing process of (**a**) PDMS-PUTI, (**b**) AgPNs-3/PDMS-PUTI, (**c**) AgPNs-6/PDMS-PUTI, and (**d**) AgPNs-9/PDMS-PUTI at 25 °C in the air (scratch thickness: ~120 μm, scratch width: ~7 μm, film thickness: ~0.7 mm).

**Figure 7 materials-17-04289-f007:**
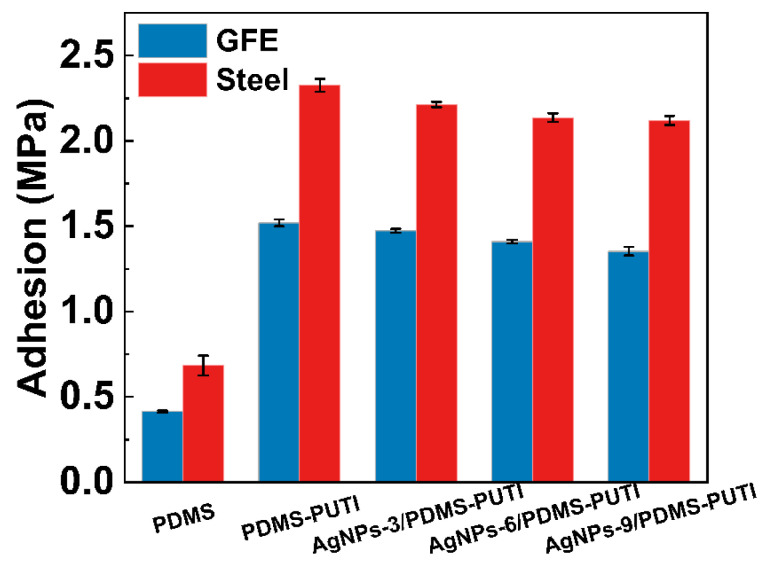
Adhesion strength of PDMS and AgNPs-x/PDMS-PUTI coatings adhered to the GFE and steel.

**Figure 8 materials-17-04289-f008:**
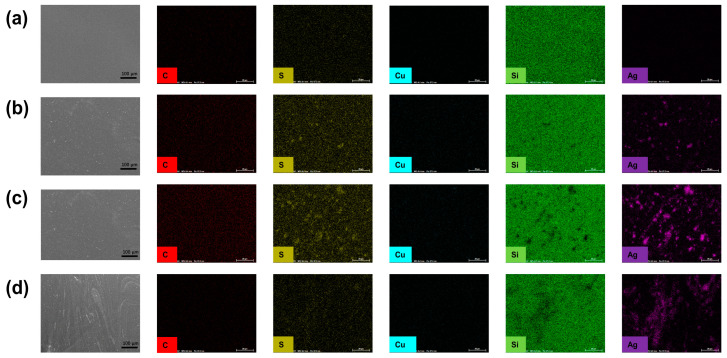
SEM and EDS spectrum images (C, S, Cu, Si, Ag) of the surface of (**a**) PDMS-PUTI, (**b**) AgNPs-3/PDMS-PUTI, (**c**) AgNPs-6/PDMS-PUTI, (**d**) AgNPs-9/PDMS-PUTI.

**Figure 9 materials-17-04289-f009:**
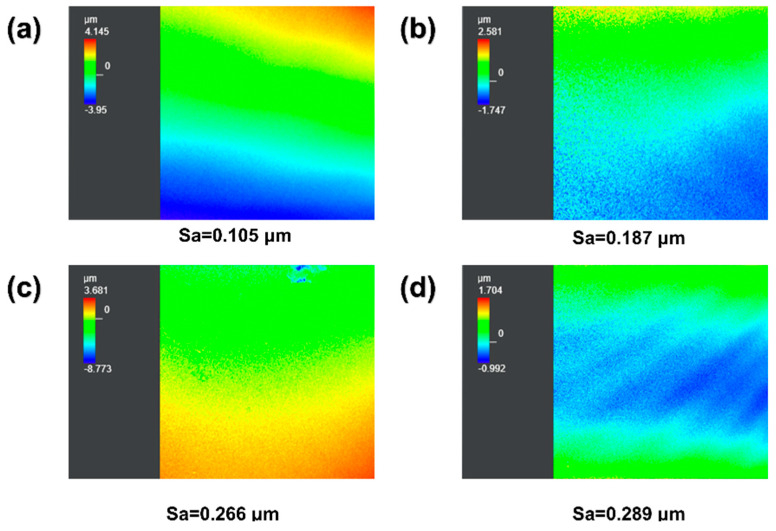
CLSM images of (**a**) PDMS-PUTI, (**b**) AgNPs-3/PDMS-PUTI, (**c**) AgNPs-3/PDMS-PUTI, and (**d**) AgNPs-3/PDMS-PUTI (image showing a 324 μm × 322 μm area).

**Figure 10 materials-17-04289-f010:**
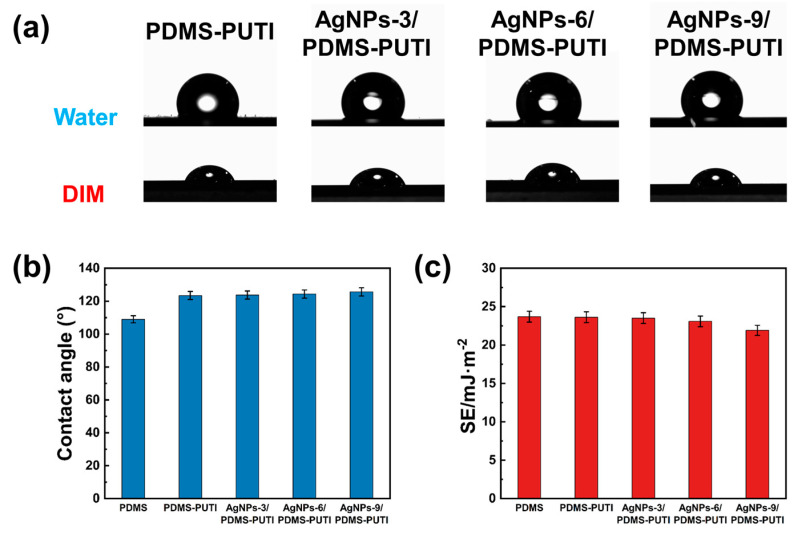
(**a**) Images of water and DIM contact angles for PDMS-PUTI, PDMS-PUTI/0.1, PDMS-PUTI/0.5, and PDMS-PUTI/1.0, (**b**) WCA of PDMS, PDMS-PUTI, PDMS-PUTI/0.1, PDMS-PUTI/0.5, and PDMS-PUTI/1.0, (**c**) SE of PDMS, PDMS-PUTI, PDMS-PUTI/0.1, PDMS-PUTI/0.5, and PDMS-PUTI/1.0.

**Figure 11 materials-17-04289-f011:**
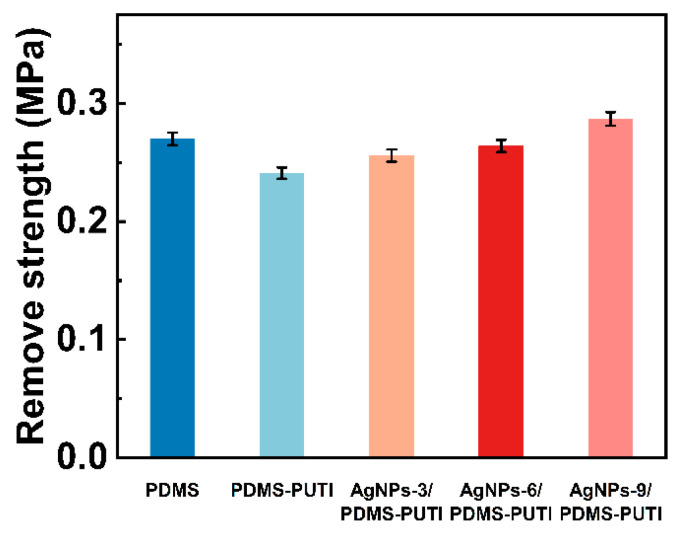
Removal strength of pseudobarnacles on each coating.

**Figure 12 materials-17-04289-f012:**
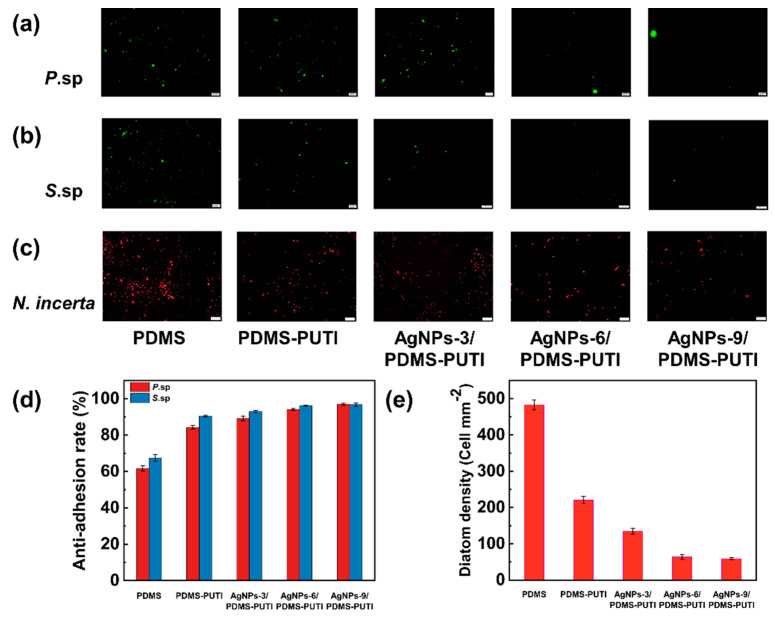
Antibacterial and anti-diatom properties of AgNPs-X/PDMS-PUTI coatings. Fluorescence images of (**a**) *P.*sp. and (**b**) *S*.sp, (**c**) *N. incerta* adhering to PDMS, PDMS-PUTI, AgNPs-3/PDMS-PUTI, AgNPs-6/PDMS-PUTI, and AgNPs-9/PDMS-PUTI, (**d**) quantitative evaluation of *P.*sp. and *S.*sp. adhesion rates, (**e**) quantitative colonization density of *N. incerta* on coating surfaces.

## Data Availability

The original contributions presented in the study are included in the article, further inquiries can be directed to the corresponding author.

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
