# Peer review of "AgNP Composite Silicone-Based Polymer Self-Healing Antifouling Coatings"

_materials, 2024, doi:10.3390/ma17174289_

Round 1
Reviewer 1 Report
Comments and Suggestions for Authors
This paper reports the preparation of silicone-based Ag nanoparticles (AgNPs)/polymer composites for anti-biofouling applications. The authors synthesized hydrogen-bonding polymers by reacting APT-PDMS with IDPI, IPAL, and CS2, prepared AgNPs composites, and examined their self-healing ability, adhesion properties, surface performance, and anti-biofouling functions. I have read this manuscript carefully and determined that it is not worthy of publication in academic journals.
First of all, the synthesized materials have not been properly characterized at all. The NMR spectrum shown in Figure 2 is too poor to be discussed. This spectrum does not indicate that the target polymer was synthesized. The analysis of the IR spectrum is also too crude. The GPC chart shown in Figure 4 indicates the formation of a polymer with Mw = 21.7 kDa. However, considering Mw of APT-PDMS is 3000 g/mol, the polymer is only a trimer of the repeating chain shown in Figure 1a. Is the polymer a self-standing solid material or viscous liquid? The authors are required to show mechanical properties and/or appearance of the polymer material. If the polymer were a viscous material with fluidity, the self-healing test would be completely meaningless.
What is the value “x” in the AgNPs-x/polymer composites? The authors state that the AgNPs are uniformly dispersed for x = 3 and 6. But I think that the dispersion of AgNPs is not uniform from Figures 5 and 7. It is obvious that PDMS-based polymers are hydrophobic and that the more AgNPs there are, the greater the antibacterial properties become, but there is no academically useful information about them.
Due to these reasons, I do not recommend publication of this manuscript.
Others:
-- Caption of Figure 5: AgPNs --> AgNPs
-- Figure 7: The letters indicating the atom species are too small.
-- Figures 2-4: The vertical axes should be appropriately labelled.
Comments on the Quality of English LanguageThe authors' English should be edited by a specialist.
Author Response
Reviewer #1:
- First of all, the synthesized materials have not been properly characterized at all. The NMR spectrum shown in Figure 2 is too poor to be discussed. This spectrum does not indicate that the target polymer was synthesized. The analysis of the IR spectrum is also too crude.
Response: Thank you for your comments and suggestions. We have made the following changes: replaced the NMR and infrared spectrum and provided a more detailed explanation of the IR spectrum.
The new additions and the modifications in “3.1. Structural characterization” are as follows:
...The peak at 1445 cm-¹ corresponds to the stretching of N-C=S. Furthermore, due to the significant molar ratio of IPDI to IPAL, the characteristic absorption peak of the C=N bond at 1650 cm-¹ is masked by the absorption peak of the C=O bond at 1630 cm-¹. This overlap results in a less distinct C=N absorption peak at 1650 cm-¹.
- The GPC chart shown in Figure 4 indicates the formation of a polymer with Mw = 21.7 kDa. However, considering Mw of APT-PDMS is 3000 g/mol, the polymer is only a trimer of the repeating chain shown in Figure 1a. Is the polymer a self-standing solid material or viscous liquid? The authors are required to show mechanical properties and/or appearance of the polymer material. If the polymer were a viscous material with fluidity, the self-healing test would be completely meaningless.
Response: Thank you very much for your review comments. We have provided a more detailed explanation in the manuscript. The polymer we synthesized is a solid coating with a certain mechanical strength. As shown in the figure 5, we have included image of the polymer drying into a film, as well as images of the mechanical tensile properties.
The new additions and the modifications in “3.1. Structural characterization” are as follows:
Figure 2 displays the 1H NMR spectrum of AgNPs-x/PDMS-PUTI. The polymer is a linear polymer with Si-O bonds as the main chain, containing urea group, imine bonds and thiourea group, where multiple strong cross-linking hydrogen bonds are formed between the polyurea and thiourea bonds, crosslinking into in an elastomer at room temperature. The proton signals of the urea linkages and adjacent methylene groups were located at 3.60 (10), 3.03 (11), and 3.14 (8) ppm, respectively.
3.What is the value “x” in the AgNPs-x/polymer composites? The authors state that the AgNPs are uniformly dispersed for x = 3 and 6. But I think that the dispersion of AgNPs is not uniform from Figures 5 and 7. It is obvious that PDMS-based polymers are hydrophobic and that the more AgNPs there are, the greater the antibacterial properties become, but there is no academically useful information about them.
Response: Thank you very much for your valuable comments and questions. “X” represents the mass fraction of AgNPs in the polymer. Figure 6 (previously Figure 5) does not show the dispersion performance of AgNPs. We replaced the SEM photos in Figure 8 (previously Figure 7). In this set of photos, it can be clearly shown that AgNPs can be uniformly dispersed when the amount of addition is low, and there is an obvious uneven distribution when the amount of addition is increased to 9 wt%.
- Caption of Figure 5: AgPNs --> AgNPs
Response: Thank you for pointing out our errors; we have made the necessary corrections.
- Figure 7: The letters indicating the atom species are too small.
Response: Thank you for pointing out our errors; we have made the necessary corrections.
- Figures 2-4: The vertical axes should be appropriately labelled.
Response: Thank you for pointing out our errors; we have made the necessary corrections.

Reviewer 2 Report
Comments and Suggestions for Authors
The authors present an approach for the development and testing of high performance silicone antifouling coatings with silver nanoparticles. This is a well organized paper. Some suggestions to make the manuscript more robust and reader friendly are described below.
Line 45 – This sentence is odd/artificial, would consider rephrasing it: The low surface free energy and elastic modulus, making it difficult for fouling organisms to adhere to the surface or allowing them to be easily removed by the shear force of water flow.
Line 121 – The significance of “x” in AgNPs-x needs to be explained in the preparation section. It is unclear if this is weight ratio or something else.
Line 191 – Consider adding a photograph of the material prepared in Section 2.2 before discussing specific properties.
Line 295 – Consider adding contact angle figure similar to Fig 9a in https://papers.ssrn.com/sol3/papers.cfm?abstract_id=4750862
Line 327 – “best” performance. By what metric? By how many percentage is it improved?
The statistical analysis in this paper is lacking. The authors say that the coating is “good” or the performance is “exceptional”. But the actual impact of the properties, variation between AgNPs- 3, 6 or 9 ratio etc. is never explained in a scientific manner. PDMS-PUTI appears to perform better than PDMS. But the addition of AgNPs only marginally improve performance? Do statistical analysis support this conclusion? What is the authors recommendation? What do the error bars signify? Consider adding some descriptive sentences to address this.
The cost aspect of Ag NP is significant. Would this be a practical option or would it be cost prohibitive? Would it work in a wider context outside a lab? Consider adding in discussion to highlight any possible limitations.
The main contribution and real life application of the material must be discussed further. Consider adding some discussion regarding the applicability and use of these materials for coating marine artificial structures (this is the primary application of the material according to the authors in the introductory section). If there are specific applications, these should be highlighted (beyond simply stating coating marine artificial structures). Can this technology be applied widely, and if so, how?
Comments on the Quality of English LanguageNo major edits.
Author Response
Responds to the reviewer’s comments:
Reviewer #2:
- The authors present an approach for the development and testing of high performance silicone antifouling coatings with silver nanoparticles. This is a well organized paper. Some suggestions to make the manuscript more robust and reader friendly are described below.
Line 45 – This sentence is odd/artificial, would consider rephrasing it: The low surface free energy and elastic modulus, making it difficult for fouling organisms to adhere to the surface or allowing them to be easily removed by the shear force of water flow.
Response: Thank you very much for your suggestions. We have made the changes according to your request.
- Line 121 – The significance of “x” in AgNPs-x needs to be explained in the preparation section. It is unclear if this is weight ratio or something else.
Response: Thank you very much for your suggestions. “X” represents the mass fraction of AgNPs in the polymer.
The final product was then subjected to rotary evaporation and vacuum drying to obtain PDMS-PUTI. 10 g of the PDMS-PUTI was subsequently dissolved in THF, and AgNPs (0.3 g,3 wt %)were added. Finally, the mixture was poured into a polytetrafluoroethylene (PTFE) mold and vacuum-dried. All prepared coatings were named AgNPs-X%/PDMS-PUTI, and X% represents the mass ratio of AgNPs.
- Line 191 – Consider adding a photograph of the material prepared in Section 2.2 before discussing specific properties.
Response: Thank you for your comments and suggestions. We added exterior photos of the coating.
- Line 295 – Consider adding contact angle figure similar to Fig 9a in https://papers.ssrn.com/sol3/papers.cfm?abstract_id=4750862
Response: Thank you very much for your suggestions. We have made the changes according to your request.
- Line 327 – “best” performance. By what metric? By how many percentage is it improved?
Response: Thank you for your comments and suggestions. The antibacterial capacity of the coatings was measured using antibacterial rates. AgNPs-9/PDMS-PUTI exhibited the highest antibacterial rates, with 97.01% against P.sp and 96.71% against S.sp, which represents an increase of 2.94% and 0.60%, respectively, compared to the antibacterial rates of AgNPs-6/PDMS-PUTI, which were 94.07% and 96.21%.
- The statistical analysis in this paper is lacking. The authors say that the coating is “good” or the performance is “exceptional”. But the actual impact of the properties, variation between AgNPs- 3, 6 or 9 ratio etc. is never explained in a scientific manner. PDMS-PUTI appears to perform better than PDMS. But the addition of AgNPs only marginally improve performance? Do statistical analysis support this conclusion? What is the authors recommendation? What do the error bars signify? Consider adding some descriptive sentences to address this.
Response: Thank you for your comments and suggestions. We have added explanations regarding the effects of increased AgNPs content on performance in the test section, including explanations for error bars and other details. We also have added a detailed description in the conclusion section.
2.7. Antibacterial and Antidiatom Settlement Tests
All media were sterilized at 121 °C for 30 minutes prior to use. Inoculation and immersion experiments were carried out in an AIRTECH clean bench after 30 minutes of UV irradiation[22]. The number of bacteria in each field of view was counted using Image J software and compared to the bacteria count on a blank glass plate to determine the antibacterial rate for different images. The average value was used as the antibacterial rate for the sample, and the error bars represent the range of antibacterial rates across different fields of view.
3.5. Antibacterial and anti-diatom properties
Among them, AgNPs-9/PDMS-PUTI had the least degree of bacterial adhesion. The surface energy of PDMS-PUTI is lower, making it easier for bacteria to detach. Additionally, the thiourea groups themselves have certain antimicrobial properties, giving PDMS-PUTI better antibacterial capabilities. Furthermore, AgNPs also slightly reduced the surface energy of the coating. The synergistic effect of AgNPs' excellent antimicrobial properties and the low surface energy performance of the polymer resulted in the coating having better antifouling properties than PDMS-PUTI. Figure 11c displays fluorescent images of samples immersed in a diatom culture medium for one day.
- The cost aspect of AgNP is significant. Would this be a practical option or would it be cost prohibitive? Would it work in a wider context outside a lab? Consider adding in discussion to highlight any possible limitations.
The main contribution and real life application of the material must be discussed further. Consider adding some discussion regarding the applicability and use of these materials for coating marine artificial structures (this is the primary application of the material according to the authors in the introductory section). If there are specific applications, these should be highlighted (beyond simply stating coating marine artificial structures). Can this technology be applied widely, and if so, how?
Response: Thank you for your comments and suggestions. We have added a detailed description in the conclusion section.
- Conclusion
The cost aspect of AgNP is significant, but by combining them with high-performance polymer matrices to create composite coatings, the amount of AgNPs can be minimized while maximizing their excellent antibacterial properties. This approach holds promise for achieving high antibacterial rates with a lower amount of AgNPs, leading to cost-effective nanocomposite polymer coatings in the future. As a low-surface-energy antifouling coating, this coating can be applied to various ships, effectively reducing the adhesion of marine organisms, thereby decreasing energy consumption and minimizing the risk of biological invasion. This work provides a promising pathway towards the development of high-performance silicone-based coatings for marine anti-biofouling.

Reviewer 3 Report
Comments and Suggestions for Authors
Manuscript ID: materials-3130337
Article
AgNPs composite silicone-based polymer self-healing antifouling coating
In this study, the authors studied and showed the synthesis of a novel silicone polymer polydimethylsiloxane (PDMS)-based poly(urea-thiourea-imine) (PDMS-PUTI) via stepwise reactions of aminopropyl-terminated polydimethylsiloxane (APT-PDMS) with isophorone diisocyanate (IPDI), isophthalaldehyde (IPAL), and carbon disulfide (CS2). Subsequently, a nanocomposite coating (AgNPs-x/PDMS-PUTI) was prepared by adding silver nanoparticles (AgNPs ) to the polymer PDMS-PUTI.
The study is critical and showed essential characteristics and analysis including NMR, SEM, EDX, FTIR and other techniques.
To make the study acceptable and cover some important areas, the following points must be solved:
1) Sec 2.6 and 2.7 need at least two references.
2) The following figures are unclear and the resolution is very bad (Figures 7 and 11(a, b, and c).
3) The conclusion is missing important sections like adhesion, roughness, and surface properties)
4) In your abstract you talked about the novelty of the study and one of that is the reduction of surface energy and enhancement of antifouling performance which was not clear in the study.
5) Can you add a figure showing enhancement of corrosion for steel coupon?
Comments on the Quality of English LanguageMinor English mistakes
Author Response
Reviewer #3:
In this study, the authors studied and showed the synthesis of a novel silicone polymer polydimethylsiloxane (PDMS)-based poly(urea-thiourea-imine) (PDMS-PUTI) via stepwise reactions of aminopropyl-terminated polydimethylsiloxane (APT-PDMS) with isophorone diisocyanate (IPDI), isophthalaldehyde (IPAL), and carbon disulfide (CS2). Subsequently, a nanocomposite coating (AgNPs-x/PDMS-PUTI) was prepared by adding silver nanoparticles (AgNPs ) to the polymer PDMS-PUTI.
The study is critical and showed essential characteristics and analysis including NMR, SEM, EDX, FTIR and other techniques.
To make the study acceptable and cover some important areas, the following points must be solved:
1)Sec 2.6 and 2.7 need at least two references
Response: Thank you for your comments and suggestions. We have added references: the measurements of contact angles and surface energy are from DOI: 10.1002/app.1969.070130815; the reference for simulated barnacle tests is 10.1016/j.cej.2020.126870; and for antibacterial and antifouling properties, the sources are 10.1002/adfm.202011145 and 10.1039/c9ta09794e.
2)The following figures are unclear and the resolution is very bad (Figures 7 and 11(a, b, and c).
Response: Thank you for your comments and suggestions. We have replaced Figures 7 and 11.
3)The conclusion is missing important sections like adhesion, roughness, and surface properties.
Response: Thank you for your comments and suggestions. We have made the following modifications in the conclusion section
- Conclusion
Besides, the incorporation of AgNPs as nanofillers into the polymer, combined with the low surface energy characteristics of silicone and the antibacterial properties of AgNPs, further optimized the fouling release and antifouling performance of AgNPs-x/PDMS-PUTI coatings. However, with the increase of AgNPs content, the adhesion of the coating decreased, the roughness increased, and the self-healing ability weakened. In conclusion, AgNPS-9 /PDMS-PUTI had better comprehensive properties. The cost aspect of AgNP is significant, but by combining them with high-performance polymer matrices to create composite coatings, the amount of AgNPs can be minimized while maximizing their excellent antibacterial properties. This approach holds promise for achieving high antibacterial rates with a lower amount of AgNPs, leading to cost-effective nanocomposite polymer coatings in the future. As a low-surface-energy antifouling coating, this coating can be applied to various ships, effectively reducing the adhesion of marine organisms, thereby decreasing energy consumption and minimizing the risk of biological invasion. This work provides a promising pathway towards the development of high-performance silicone-based coatings for marine anti-biofouling.
4) In your abstract you talked about the novelty of the study and one of that is the reduction of surface energy and enhancement of antifouling performance which was not clear in the study.
Response: Thank you for your comments and suggestions. We added the following explanation to the antibacterial testing section.
3.5. Antibacterial and anti-diatom properties
AgNPs contents [34]. Among them, AgNPs-9/PDMS-PUTI had the least degree of bacterial adhesion. The surface energy of PDMS-PUTI is lower, making it easier for bacteria to detach. Additionally, the thiourea groups themselves have certain antimicrobial properties, giving PDMS-PUTI better antibacterial capabilities. Furthermore, AgNPs also slightly reduced the surface energy of the coating. The synergistic effect of AgNPs' excellent antimicrobial properties and the low surface energy performance of the polymer resulted in the coating having better antifouling properties than PDMS-PUTI. Figure 11c displays fluorescent images of samples immersed in a diatom culture medium for one day.
5) Can you add a figure showing enhancement of corrosion for steel coupon?
Thank you for your comments and suggestions. In practical applications, low surface energy marine antifouling coatings require the synergistic use of primer, tie-coat, and topcoat. Our research primarily focuses on the topcoat, with the main objective of enhancing its marine antifouling capability. We did not design and test the materials specifically for corrosion resistance.

Round 2
Reviewer 2 Report
Comments and Suggestions for Authors
The authors have made several updates that have improved the paper.
Reviewer 3 Report
Comments and Suggestions for Authors
Many thanks for improving the manuscript